# Acquired Hemophilia A after SARS-CoV-2 Infection: A Case Report and an Updated Systematic Review

**DOI:** 10.3390/biomedicines11092400

**Published:** 2023-08-28

**Authors:** Márton Németh, Diána Mühl, Csaba Csontos, Ágnes Nagy, Hussain Alizadeh, Zsolt Szakács

**Affiliations:** 1Department of Anesthesiology and Intensive Therapy, Medical School, University of Pécs, 7624 Pécs, Hungary; nemethmarton85@gmail.com (M.N.); muhl.diana@pte.hu (D.M.); csontos.csaba@pte.hu (C.C.); 2First Department of Medicine, Medical School, University of Pécs, 7624 Pécs, Hungary; m.nagy.agnes@pte.hu (Á.N.); szakacs.zsolt@pte.hu (Z.S.)

**Keywords:** acquired hemophilia, SARS-CoV-2, COVID-19, hemostasis, coagulopathy, case report

## Abstract

The role of severe acute respiratory syndrome coronavirus-2 (SARS-CoV-2) has been implicated in the pathogenesis of acquired hemophilia A (AHA). The aim of this study is to report our case and to summarize clinical studies on de novo AHA after SARS-CoV-2 infection. We performed a systematic search on the association of SARS-CoV-2 with AHA in four medical databases up to 28 May 2023. Eligible studies should include de novo AHA patients who had SARS-CoV-2 infection before or concomitant with the diagnosis of AHA. Findings were synthesized narratively. In addition, we report the case of a 62-year-old female patient, who presented to our clinic with left flank pain 2 weeks after SARS-CoV-2 infection. Clinical investigations confirmed AHA and imaging studies revealed retroperitoneal bleeding. Her hemostasis was successfully secured with bypassing agents; however, despite immunosuppressive therapy, high inhibitor titer persisted. In the systematic review, we identified only 12 relevant cases with a questionable cause–effect relationship between SARS-CoV-2 infection and AHA. Based on the qualitative analysis of the relevant publications, current clinical evidence is insufficient to support a cause–effect relationship. The analysis of data from ongoing AHA registries can serve further evidence.

## 1. Introduction

Acquired hemophilia A (AHA) is an immune-mediated condition, in which neutralizing autoantibodies (i.e., inhibitors) decrease coagulation factor VIII (FVIII) activity [1]. AHA is a rare disease with an incidence of around 1–2 per million annually [2,3]. The disease usually tends to manifest itself in late-middle-aged and elderly subjects, but a postpartum incidence peak has been reported as well. AHA is a life-threatening condition with an anticipated mortality reaching 30% [4]. The most up-to-date international guidelines on diagnostics and therapeutics of AHA were released in 2020 [5]. A recent, thorough review about diagnostics and treatment options of AHA was published by Zanon et al. in 2023 [6].

Provoking or precipitating factors, most commonly, malignancies, autoimmune diseases, infections, and pregnancy, can be identified in approximately 50% of cases [4]. The role of severe acute respiratory syndrome coronavirus-2 (SARS-CoV-2) with the corresponding clinical syndrome, coronavirus diseases 2019 (COVID19), and vaccination against SARS-CoV-2 have been implicated in the pathogenesis as well [7].

The aim of this study is twofold: (1) to report a case of de novo AHA after SARS-CoV-2 infection, and (2) to provide an updated summary on human clinical studies that reported on de novo AHA after SARS-CoV-2 infection.

## 2. Materials and Methods

### 2.1. Case Presentation

This study includes a case report that matches the question of the systematic review (see below). The subject of the case report was treated in our institution (University of Pécs, Pécs, Hungary) and provided written consent for publishing her anonymized data.

### 2.2. Systematic Review

The a-priori protocol of the systematic review is available in the International Prospective Register of Systematic Reviews (PROSPERO) under registration number CRD42023422263. No protocol amendments were made.

A systematic search was carried out in MEDLINE (via PubMed), EMBASE, Scopus, and Cochrane Trials on 28 May 2023 with the query “(COVID-19 OR SARS-CoV-2 OR coronavirus) AND hemophilia”, without field restrictions. Additional records were collected from the cited and citing articles of papers assessed by full-text during selection. All records were combined in a reference manager software (EndNote version x9 for Windows, Clarivate, UK); then, 2 review authors performed 2-step selection (by title and abstract, then by full-text) in duplicate. Discrepancies were resolved by third party arbitration.

To be eligible for systematic review, a study should include newly diagnosed (de novo) AHA patients who were confirmed to be positive for SARS-CoV-2 before or at the same time of the diagnosis of AHA. No other restrictions were applied.

The following data were collected from the eligible papers: demography (country, age, and sex), medical history focusing on co-morbidities, AHA-related data (type of bleeding, diagnostic work-up including FVIII activity and inhibitor titer, treatment including hemostatics and immunomodulators, disease outcome including thrombotic complications), and SARS-CoV-2-related data (diagnostics, severity, timing of the diagnosis in relation to the diagnosis of AHA).

As only case reports were found to be eligible, data were summarized in a systematic review.

## 3. Results

### 3.1. Case Presentation

A 62-year-old woman with a history of warm antibody autoimmune hemolytic anemia and suspected systemic lupus erythematosus originally diagnosed in January 2013, unvaccinated against SARS-CoV-2, had polymerase chain reaction (PCR)-verified, mild COVID-19 that needed no antiviral treatment, anticoagulation, or hospitalization. Since the beginning of the COVID pandemic, she received no immunosuppressive therapy and her AIHA did not require red blood cell transfusion or any other systemic therapy.

Two weeks later, on index admission, she presented to the emergency department after a fall at home and complaints of fatigue and severe pain in the pelvic and left flank areas. She tested negative for SARS-CoV-2 both with rapid antigen test (RAT) and PCR. The patient denied taking any antithrombotic agent. Initial laboratory findings showed normocytic normochromic anemia (hemoglobin: 85 g/L), normal platelet count, normal prothrombin time (PT, 12 s, reference: 9–12 s), and normal fibrinogen level; activated partial thromboplastin time (APTT) was not requested. Transabdominal ultrasound suggested a cystic retroperitoneal lesion, most probably of bleeding origin, and a contrast-enhanced computed tomography (CT) confirmed the diagnosis of retroperitoneal hematoma (Figure 1).

After a multidisciplinary team discussion, urgent surgical intervention was recommended and it revealed large retroperitoneal hematoma without an obvious source of bleeding. After thorough hematoma evacuation, surgical swabs were left in situ to tamponade the bleeding site. Due to the continuous, oozing bleeding, the patient received hemosubstitution as per the massive transfusion protocol. The postoperative hemostatic assessment showed an isolated prolongation of APTT. Recombinant activated factor VII (rFVIIa, eptacog alpha, NovoSeven, Novo Nordisk Inc.) was given to secure hemostasis. Shortly after surgical intervention, the patient became hemodynamically unstable and was admitted to the intensive care unit (ICU).

We re-assessed the patient’s hemostasis with a viscoleastic test, ClotPro (Diacare Solution, Hungary), which showed a prolonged clotting time with the IN-test, representing the activity of the intrinsic coagulation pathway. On the EX-test, beside normal coagulation time, an adequate maximal clot firmness was detected without hyperfibrinolysis (Figure 2). There was no definitive, surgically manageable source of bleeding. To prevent further blood loss, rFVIIa was given intravenously (IV) 90 μg/kg body weight every 3 h, which, together with other supportive measures, resulted on establishment of stable hemostasis. The patient’s fluid replacement was based on goal-directed hemodynamic endpoints and the blood products were substituted according to physiologic transfusion triggers such as central venous oxygen saturation, lactate clearance, and carbon dioxide gap.

In the meantime, a detailed hemostatic evaluation was performed. APTT mixing studies showed no correction of prolonged APTT (the lupus anticoagulant test was negative), FVIII activity was unmeasurably low (<0.1%), factor IX activity was moderately low (6.9%), and the modified Bethesda test showed high titer of inhibitory autoantibodies (2283 BU). These results confirmed the diagnosis of AHA.

To secure hemostasis, we continued the rFVIIa treatment according to the recommended dosing schedule and we started the patient on immunosuppressive therapy (IST) for the purpose of eradicating the indeterminable high inhibitory antibodies. Initially, the patient received 250 mg methylprednisolone IV. After multiple transfusions, the patient developed transfusion-related acute lung injury that progressed to acute respiratory distress syndrome, requiring endotracheal intubation and invasive mechanical ventilation. To manage severe hypoxia, airway pressure release ventilation was commenced and lung protective ventilator settings were used to avoid ventilator-associated lung injury. For circulatory support, low-dose norepinephrine was initiated to maintain adequate blood pressure. Because of mildly elevated procalcitonin and C-reactive protein, empirical broad spectrum antibiotic therapy was initiated.

When the patient’s condition stabilized, we continued the inhibitor eradication with methylprednisolone and cyclophosphamide as per the institutional recommended dosing protocol. The bleeding tendency reduced; however, the titer of inhibitors remained high. Due to persistently high inhibitor titer and sepsis, we indicated plasmapheresis which resulted in rapid, but short-lasting reduction in the titer of inhibitory antibodies. Retroperitoneal swabs were regularly changed with prophylactic pre-procedural rFVIIa administration. During the surgical control of the retroperitoneal bleeding, the severity of bleeding, and the amount of blood clot decreased gradually. The patient’s general condition and the manifested bleeding improved with supportive treatment and regular administration of rFVIIa, and she did not require blood product transfusion. However, the FVIII level remained extremely low (<0.1%) with no significant change in the inhibitor titer (>300 BU).

To be able to wean the patient gradually from the mechanical ventilator, percutaneous tracheostomy was performed after the administration of an additional dose of rFVIIa product prior to the procedure using an atraumatic Blue Rhino^®^ G2-Multi Percutaneous Tracheostomy set. After 19 days of mechanical ventilation, the patient was gradually weaned from the ventilator and decannulation was performed prior to discharge. The mobilization and rehabilitation could only take place in small steps due to the critical illness polyneuropathy and myopathy. After cardiopulmonary stabilization, musculoskeletal mobilization, and rehabilitation, the patient was transferred to the hematology division. During the ICU stay, she received a total of 34 packs of red blood cells, 20 units of fresh-frozen plasma, 20 packs of pooled platelets, and 759 mg rFVIIa.

To maintain hemostasis, rFVIIa was continued, which was subsequently switched to factor VIII inhibitor bypass activity (FEIBA 3 × 5000 U/week) due to logistic issues. Because of a persistently high titer of inhibitory autoantibodies, we switched the IST to the following protocol: 1000 mg cyclophosphamide IV on days 1 and 22; 125 mg methylprednisolone IV on days 1, 8, 15, and 22; and 375 mg/m^2^ mg rituximab IV on days 1 and 22. Despite being bleeding-free, the FVIII activity and inhibitor level did not reduce significantly. The patient refused the inpatient rehabilitation; therefore, after 6 weeks of in-hospital rehabilitation and mobilization, she was discharged from the hospital. Potential provoking factors of AHA were retrospectively assessed; apart from SARS-CoV-2 positivity, no other provoking infections were identified with serology. Detailed imaging and endoscopic and serologic studies ruled out malignancies.

FEIBA prophylaxis was continued on an ambulatory basis, and she remained on low-dose methylprednisolone (4 mg/d per os) as per the recommendation of the immunology department.

At +4 month, the patient developed traumatic right knee peri-articular inflammation, which required surgical intervention with peri-procedural rFVIIa treatment. The procedure went smoothly without any complication after the effective management of the patient’s hemostasis. A detailed ambulatory immunological assessment was performed, which, considering the patient’s history (AIHA, lupus-like conditions) and the results of immune serology and immune flow cytometric tests, established the diagnosis of mixed connective tissue disease, and we recommended methotrexate treatment (oral, 15 mg/week). However, as the patient developed severe neutropenia, methotrexate was stopped. It is important to note that the results of the immunological tests should be treated cautiously because the patient had been treated with long-term glucocorticoid therapy, increasing the rate of false negativity of the laboratory findings.

At +6 month, the patient suffered a traumatic wrist injury without intraarticular bleeding, requiring hospital admission and minor surgical intervention with peri-procedural rFVIIa treatment. FVIII activity did not recover (<0.1%), the inhibitor level remained high (3614 BU), and the IN-test on ClotPro showed a prolonged clotting time (917 s, reference: 139–187 s). Due to the persistent high inhibitor titer, a new cycle of combined immunosuppressive therapy was started as per the following IST protocol for AHA: 1000 mg cyclophosphamide IV on days 1 and 22; 40 mg dexamethasone orally on days 1, 8, 15, and 22; and 100 mg rituximab IV on days 1, 8, 15, and 22.

Last seen at +8 month, the patient was self-sustaining and capable of walking without any difficulties. Until now, no clinically relevant thrombotic complication has been documented.

Timeline of the case is summarized in Figure 3.

### 3.2. Systematic Review

#### 3.2.1. Search and Selection

A flowchart of the search and selection is summarized in Appendix A. After selection by title and abstract, 18 records were eligible for inclusion, of which, 3 were review articles [7,8,9] and another 3 reported about AHA relapse during or after SARS-CoV-2 infection [10,11,12]. Finally, 12 studies (11 full-text articles and 1 conference abstract) were included in the systematic review (Table 1) [13,14,15,16,17,18,19,20,21,22,23,24].

#### 3.2.2. Demography

Out of the 12 studies, 5 reported cases from the US, 1 study from Iran, and all the other from Europe (Austria, Serbia, Portugal, Italy, Croatia, and Poland; one case each). All but one study reported on middle-aged or elderly patients, and seven of twelve patients were male.

#### 3.2.3. Diagnostics and Severity of SARS-CoV-2 Infection

Surprisingly, only four studies reported that the patient was positive for SARS-CoV-2 with PCR; none of the others reported on diagnostic procedures or used serological tests (no studies reported the use of rapid antigen test, RAT). The reported severity of the infection was severe/critical in five cases and asymptomatic/mild in six cases (unknown in one study).

#### 3.2.4. Diagnostics and Severity of Acquired Hemophilia A

Cutaneous/subcutaneous and muscle bleedings were the dominant types of bleedings. Other sites of bleeding included urinary (two cases), conjunctival (one case), airways or lung (three cases), postpartum vaginal (one case), and puncture sites (two cases). In 10 studies, the FVIII activity was unmeasurably low, 1 study reported only normal FVIII activity after FVIII treatment, and another did not report it at all. Only four studies reported the methodology of FVIII and inhibitor measurements. The highest anti-FVIII titer was 2222 BU and the lowest was 21 BU. In one study, the inhibitor titer was probably not measured.

#### 3.2.5. Treatment of Acquired Hemophilia A

For hemostatic control and bleeding prevention, rFVIIa was used most frequently (six studies), followed by FEIBA (four studies) and recombinant porcine factor VIII (rpFVIII, two studies). In one study, the patient received emicizumab in the acute phase. In two studies, no agents were used for securing hemostasis.

For inhibitor eradication, nine studies used glucocorticoids and six used rituximab and cyclophosphamide alone or concomitantly.

#### 3.2.6. Outcome

Three studies reported a fatal outcome (one case of disseminated intravascular coagulation, two cases of infection); all the others reported a positive study outcome (mostly remission of AHA). Apart from the single disseminated coagulopathy case, no thrombotic events were registered during the treatment of AHA.

#### 3.2.7. Relationship of SARS-CoV-2 Infection and Acquired Hemophilia A

The exact temporal relationship of the conditions was very hard to define and remained unknown in two studies. In all the other studies, AHA was diagnosed around the time of the diagnosis of SARS-CoV-2 infection or within 4 months after the infection.

Provoking factors other than SARS-CoV-2 infection were reported in four studies (two cases of prior malignancies, one case was post-partum, and another case had prior viral infection and autoimmune thyroiditis). Of note, most studies did not detail the investigations for other potential provoking factors.

## 4. Discussion

In this study, we aimed to summarize clinical information about AHA developed after SARS-CoV-2 infection. Our updated systematic search identified 12 relevant case reports (Table 1), but failed to identify studies of other, more complex designs. We report our own case of AHA, which developed after SARS-CoV-2 infection.

AHA, as with many other immune-mediated conditions, tends to be associated with other immune-mediated conditions. In the EACH2 study and in the HTRS Registry, 27% and 28.4% of AHA patients had an associated autoimmune diagnosis, respectively [4,25]. Among these, rheumatoid arthritis and systemic lupus erythematosus can be highlighted [25]. Note that autoimmune conditions without overt AHA can affect hemostasis too [26].

Infections can often be associated with immune dysregulation and can affect hemostasis as well. In the pathophysiology of SARS-CoV-2-infection-associated critical illness, the complex interaction of hemostasis, the innate immune system, and the excessive cytokine storm, in addition to the endothelial dysfunction, play a role in the development of multiorgan involvement and dysfunction [27]. The most often affected extrapulmonary organs are the brain, kidneys, and the liver [28]. Depending on the severity of inflammation, the cytokine storm is mirrored by the changes of inflammatory markers, such as C-reactive protein (CRP), interlekuin-6, lymphocyte count, lactate dehydrogenase (LDH), and ferritin levels [29]. Nopp et al. reported the prevalence of venous thrombosis as 7.9% in non-ICU patients and 22.7% in ICU patients. The prevalence of pulmonary embolism in non-ICU and ICU patients was reported as 3.5% and 13.7%, respectively [30]. Macrothrombosis can manifest as either pulmonary embolism or could be associated with deep venous thrombus. Microthrombosis manifests with abnormal laboratory findings such as prolonged coagulation times, moderately decreased fibrinogen, and an increased D-dimer level, a marker of fibrin degradation product and the clinical picture of acute respiratory distress syndrome [31]. With the increased rate and severity of thrombosis, there is a tendency of elevated morbidity and mortality [32]. SARS-CoV-2 infection leads in multiple pathophysiological ways to increased thrombus formation. Vascular dysfunction can occur directly through virus invasion via angiotensin-converting enzyme 2 receptor to the endothelial cells or indirectly through the cytokine storm. Endothelial damage impairs antithrombotic function with the imbalanced release of tissue factor and tissue factor pathway inhibitor [33], in addition to the production and release of von Willebrand Factor and cleavage of ultra large von Willebrand Factor by ADAMTS13 [34]. The production and release of tissue plasminogen activator (t-PA) and the decreased production of plasminogen activator inhibitor (PAI-1) lead to impaired fibrinolysis [35]. With the disruption of the endothelial glycocalyx layer, the loss of vasodilation and vasoconstriction, together with the microvascular clot formation, leads to impaired organ perfusion and to organ dysfunction and multiple organ failure [35].

Although thrombosis is more frequent, among critically ill patients treated for SARS-CoV-2 infection, few subjects have an increased risk of major bleeding. Patients with major hemorrhage had higher D-dimer, higher fibrin degradation product (FDP), lower fibrinogen, and longer PT and APTT. According to the International Society on Thrombosis and Haemostasis (ISTH) grading scores, 71% of non-survivors met the criteria of disseminated coagulation score in the later time course of diseases [36]. Therapeutic interventions to prevent thrombus formation or to manage already in situ thrombus with anticoagulants, antiplatelets, and thrombolytics further increase the iatrogenic risk of major hemorrhage. In a meta-analysis by Valeriani et al., venous thromboembolism occurred in 2.9% of patients on high-dose and in 5.7% of patients on low-dose thromboprophylaxis, and major bleeding occurred in 2.5% and 1.4% of patients on high-dose and low-dose thromboprophylaxis, respectively [37]. The administration of platelet aggregation inhibitor acetylsalicylic acid (ASA) seems to be reasonable because of its pleiotropic effects, although its effect on mortality is controversial; however, the administration of ASA seems to be safe and does not increase the occurrence of bleeding complications [38]. As fibrin breakdown is often suppressed by patients treated with COVID-19 associated with severe acute respiratory distress syndrome, therapeutic fibrinolysis seems to be beneficial in some groups of patients [39]. Patients with massive pulmonary embolism, hemodynamic instability, and severe acute right ventricular failure can benefit from thrombolytic therapy [40]. There are only some case reports where patients with microthrombosis-associated acute respiratory distress syndrome and right ventricular failure benefited from low-dose systemic thrombolytic therapy [41,42]. The pathophysiological background of AHA-caused bleeding diathesis differs from patients treated with COVID-19. In theory, the clasp between SARS-CoV-2 infection and AHA can be the potent immune activator function of the virus, as both the innate and the adaptive immune systems are affected [43,44]. The production of autoantibodies is frequently involved in the pathophysiology of SARS-CoV-2 infection, and higher titers are associated with a more severe clinical course and prognosticate poor outcomes [45]. In a retrospective cohort study among 3,814,479 subjects, the COVID-19 cohort exhibited a significantly higher risk for the de novo development of systemic lupus erythematosus, Sjögren diseases, vasculitis, and other autoimmune diseases [46]. Some authors detected a signal between COVID-19 vaccination and AHA. However, these theories have very weak evidence and the authors conclude that they cannot rule out detection bias [47].

If there is a cause–effect relationship between SARS-CoV-2 infection and AHA, if the infection is part of the multi-hits required for autoantibody formation against FVIII, or if the co-occurrence of the conditions is just incidental have not proven yet. From a mechanistic perspective, AHA fits the theory of SARS-CoV-2-induced autoimmunity. However, several concerns should be raised when reading through the reports of 13 cases (see Table 1, plus our own case).

First, in some of these reports, other triggers (malignancies, co-occurring immune-mediated conditions, viral infections other than SARS-CoV-2) were reported. Though some of the papers declared that a thorough investigation was carried out for potential provoking or precipitating factors, some failed to report the details or did not report about the investigations at all. Moreover, AHA can be the first presenting symptom of diseases being in a subclinical phase at the time of AHA diagnosis.

Second, the majority of the world’s population has already gone through either a symptomatic or an asymptomatic SARS-CoV-2 infection. The temporal relationship of the two conditions is very hard to be precisely defined. To raise a cause–effect relationship, the infection must precede the precipitated conditions; however, the required in-between time, in which we should raise a relationship between the conditions, is still unknown. Theoretically, now, almost all patients having de novo AHA must have gone through SARS-CoV-2 infection in the past.

Third, patients infected with SARS-CoV-2 are more likely to visit healthcare facilities, which increases the probability of the detection of minor bleedings and performing hemostatic laboratory tests showing an increased APTT incidentally.

Fourth, pre-existing SARS-CoV-2 infection can be questioned in some studies because many of them did not report the methodology of SARS-CoV-2 detection or used serology (and not PCR or RAT), which refers only to an earlier SARS-CoV-2 infection, almost irrespective of timing.

Fifth, case reports are particularly vulnerable to reporting bias.

## 5. Conclusions

Although from a pathophysiological perspective, there may be a relationship between SARS-CoV-2 infection or anti-COVID-19 vaccination and AHA, based on the qualitative analysis of the relevant publications, current clinical evidence is insufficient to support a cause–effect relationship between SARS-CoV-2 infection and AHA. The possible association of SARS-CoV-2 infection and acquired hemophilia is quite interesting, although not surprising, considering the known immune dysregulation following SARS-CoV-2 infection. An additional pathophysiological mechanism of SARS-CoV-2 infection-triggered autoimmunity lies in the activation of quiescent autoreactive T and B cells, as well as molecular mimicry. An analysis of data from ongoing AHA registries can provide further evidence on this matter. In addition, our case report calls attention to the importance of assessing hemostasis globally, including platelet count, PT, APTT, and fibrinogen (and even viscoelastic tests if needed) in acute bleedings and peri-procedurally.

## Figures and Tables

**Figure 1 biomedicines-11-02400-f001:**
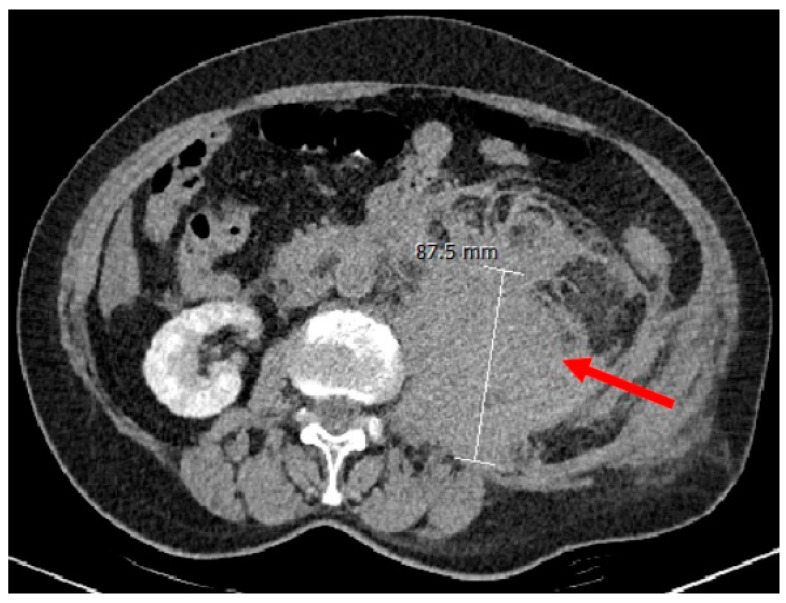
Abdominal CT angiography. CT scans revealed a retroperitoneal hematoma with an extension of 9 × 20 cm at the psoas muscle on the left side. The red arrow shows the retroperitoneal hematoma.

**Figure 2 biomedicines-11-02400-f002:**
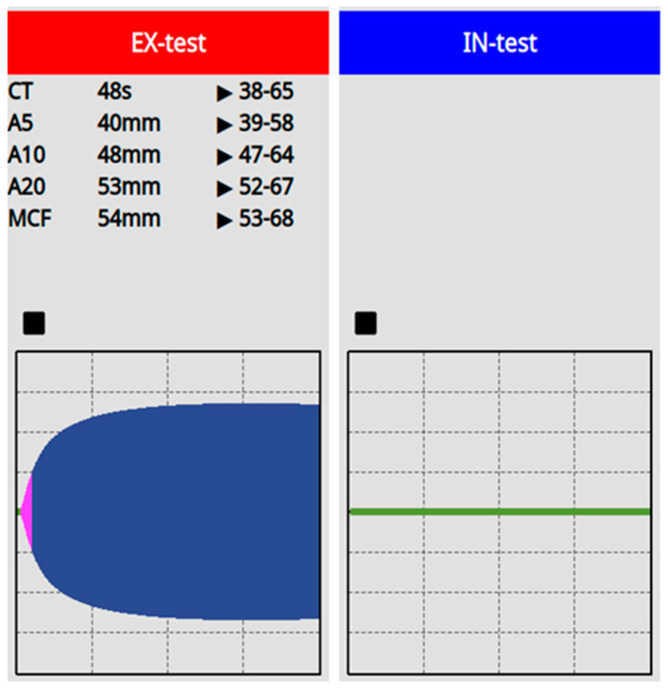
ClotPro results. The bedside test was performed post-operatively immediately on intensive care unit admission. The EX–test yielded normal coagulation time with an acceptable clot formation without any considerable fibrinolysis, whereas the IN-test shows no coagulation.

**Figure 3 biomedicines-11-02400-f003:**
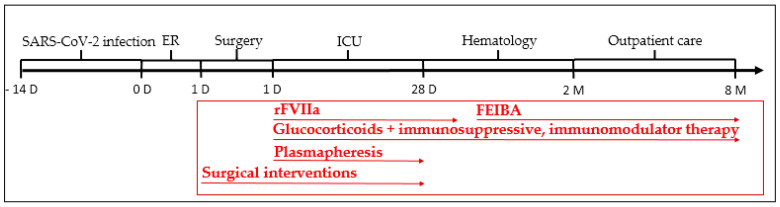
Timeline. Two weeks prior to hospital admission, the patient had SARS-CoV-2 infection. At the current emergency room (ER) presentation, the patient already had a negative test for SARS-CoV-2 virus infection. After surgical evacuation of the retroperitoneal hematoma, the patient was transferred to the intensive care unit, where hemostasis was supported by recombinant activated factor VII, and autoantibody eradication was treated with immunosuppressive therapy and plasmapheresis. The surgical hematoma was evacuated, and surgical swabs were regularly replaced. ER: emergency department, FEIBA: factor VIII inhibitor bypass activity, ICU: intensive care unit, rFVIIa: recombinant activated factor VII.

**Table 1 biomedicines-11-02400-t001:** Studies reporting on de novo acquired hemophilia A.

Study	Country	Age, Sex	Bleeding Type		SARS-CoV-2		AHA							Estimated Temporary Relationship between SARS-CoV-2 and AHA	Outcome
					Severity SARS-CoV-2 Infection	Diagnostics	FVIII Activity at Diagnosis	Method of Measurement of FVIII Activity	Inhibitor Titer at Diagnosis (BU)	Method of Measurement of Inhibitor Titer	Hemostatic Control and Bleeding Prevention	Inhibitor Eradication	Potential Provoking Factors Other than SARS-CoV-2		
Gelbenegger et al., 2022 [13]	Austria	75, M	major bleeding of the psoas muscle, pectineus muscle bleeding		critical	PCR	<1%	one-stage assay, chromogenic assays ^a^	41.8	Nijmegen-modified Bethesda assay ^b^	rFVIIa, emicizumab	prednisone, dexamethasone, rituximab	none	concomitant	AHA remission, alive at 110 days
Ghafouri et al., 2020 [14]	USA	89, M	moderate gross hematuria		critical	PCR	<1%	one-stage assay, chromogenic assays	2222	ELISA assay	rFVIIa	probably none	advanced prostate cancer in remission	concomitant	died in acute DIC
Guerra et al., 2022 [15]	USA	74, F	gross hematuria		mild	not reported	<1%	not reported	48	not reported	FEIBA, rFVIIa	prednisone, cyclophosphamide, rituximab	none	AHA 1 month after SARS-CoV-2 infection	AHA remission, discharge
Hafzah et al., 2021 [16]	USA	73, M	ecchymosis of the left thigh, arms, bruises on the trunk, conjunctival bleeding		severe	not reported	<1%	not reported	70.4	Bethesda assay	none	prednisone, cyclophosphamide	none	AHA 4 month after SARS-CoV-2 infection	AHA remission, discharge
Hajigholami et al., 2021 [17]	Iran	45, F	hemoptysis		critical	PCR	76% (after FVIII treatment)	not reported	probably not measured	not reported	FVIII, rFVIIa	none	epiglottis tumor removed 10 days before presentation	concomitant	died probably in sepsis
Lackovic et al., 2023 (conference abstract) [18]	Serbia	19, F	postpartum vaginal bleeding		not reported	not reported	not reported	not reported	not reported	not reported	unclear	unclear	pregnancy (postpartum 6th week)	concomitant	positive response to treatment
Medeiros et al., 2022 [19]	Portugal	73, M	quadriceps muscle hematoma, bleeding from puncture sites, alveolar hemorrhage, hemarthrosis		mild	IgG SARS-CoV-2 antibody	<1%	not reported	719	not reported	rFVIIa	prednisolone, rituximab	none	AHA 3 month after SARS-CoV-2 infection	death due to infection
Nardella et al., 2022 [20]	Italy	53, F	bilateral ecchymosis in the deltoid area, subcutaneous hematomas in the lower limbs, iliopsoas hematoma		mild	anti-spike protein antibody (no vaccination)	<1%	not reported	142	not reported	rFVIIa, rpFVIII, FEIBA	methylprednisolone, cyclophosphamide, rituximab, dexamethasone	none	unknown	AHA remission at 6 months
Nikolina et al., 2022 [21]	Croatia	73, M	several skin suffusions		severe or critical	PCR	<2%	clot-based method	not reported	not reported	FEIBA	methylprednisolone, cyclophosphamide	none	AHA 3 month after SARS-CoV-2 infection	AHA remission at 3 months
Olsen et al., 2021 [22]	USA	83, F	extensive ecchymoses, iliac muscle hematoma		mild	IgG SARS-CoV-2 antibody	2.2% (<10%) ^c^	one-stage assay, chromogenic assay	25	not reported	none	corticosteroids (not specified), rituximab	none	concomitant	AHA remission at 1 month
Wang et al., 2021 [23]	USA	65, M	severe subcutaneous bleedings, bleeding from puncture site, small pulmonary hematoma		asymptomatic	SARS-CoV-2 antibodies	<1%	not reported	176	Bethesda assay	rFVIIa, FEIBA	methylprednisolone, cyclophosphamide, rituximab	Hashimoto-thyroiditis, prior viral infection	unknown	AHA remission at 3 weeks
Witkowski et al., 2022 [24]	Poland	86, M	extensive cutaneous petechiae		mild	not reported	<1%	not reported	21	not reported	rFVIIa, rpFVIII	prednisone, cyclophosphamide	none	AHA 3 month after SARS-CoV-2 infection	AHA remission, discharge

^a^ After emicizumab treatment. ^b^ Modified after emicizumab treatment. ^c^ Value in parentheses concerns the result of the chromogenic assay. AHA: acquired hemophilia A. FEIBA: factor VIII inhibitor bypass activity. FVIII: factor VIII. PCR: polymerase chain reaction. SARS-CoV-2: severe acute respiratory syndrome coronavirus-2. rFVIIa: recombinant activated factor VII. rpFVIII: recombinant porcine factor VII.

## Data Availability

All data are available in the paper.

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
