# Peer review of "Acquired Hemophilia A after SARS-CoV-2 Infection: A Case Report and an Updated Systematic Review"

_biomedicines, 2023, doi:10.3390/biomedicines11092400_

Round 1
Reviewer 1 Report
Interesting case, quit heavy!
Here are my suggestions
Line 97-98: "activated partial thromboplastin time (APTT) was not measured". It is surprising that aPTT was not performed on admission, is this test not in the routine lab in your Institution? It seems important in the presence of a bleeding history: this has further delayed the final diagnosis cfr Line 104"large retroperitoneal hematoma without an obvious source of bleeding”.
aPTT is also important because a surgical intervention was necessary: isn't it part of pre-operative test?
As the cause of bleeding was not known yet, it is surprising to me that your first line of treatment was FVII an expensive factor
These points should be further discussed
Line 112: What bring the CloPro test in this case? Is it part of your routine tests?
Line 193-204 it is not clear to me if those lines are the legend of figure 3.
Conclusion for this case in particular is not discussed. Is it likely that SARSCov2 infection may have act as an additional trigger in this patient with a so heavy autoimmune clinical history?
Line 267"Note that autoimmune conditions without over AHA can affect hemostasis, too" over should be replaced by overt?
Author Response
Dear Sir/Madam,
Thank you for your proper and detailed review. I hope that our answers and sufficient and valuable for you.
1. Line 97-98: "activated partial thromboplastin time (APTT) was not measured". It is surprising that aPTT was not performed on admission, is this test not in the routine lab in your Institution? It seems important in the presence of a bleeding history: this has further delayed the final diagnosis cfr Line 104"large retroperitoneal hematoma without an obvious source of bleeding”.
aPTT is also important because a surgical intervention was necessary: isn't it part of pre-operative test?
1. Response: Thanks for pointing out this important issue. Unfortunately, the APTT was not part of the basic hemostasis panel either in the ED or as part of the pre-operative setting. But after this and similar cases, the APTT, TT, and fibrinogen have been added to the original hemostasis panel which consisted of PT (or INR) in addition to the complete blood count with differentials.
2. As the cause of bleeding was not known yet, it is surprising to me that your first line of treatment was FVII an expensive factor.
2. Response: Thank you for this question. Based on the ISTH guidelines, the first line treatment of patients with acquired hemophilia and high inhibitor titer (titers > 20 units) is rFVIIa. In case of low titer, FEIBA or other hemostatic products (e.g.; aPCC) can be used.
3. Line 112: What bring the CloPro test in this case? Is it part of your routine tests?
3.Response: ClotPro is routinely requested by the pre-operative team if there is any issue with the hemostasis or if the patient is on antithrombotic drugs.
4. Line 193-204 it is not clear to me if those lines are the legend of Figure 3.
4. Response: Thanks for bringing this mistake to our attention. The test has been corrected in the revised manuscript.
5. Conclusion for this case in particular is not discussed. Is it likely that SARSCov2 infection may have act as an additional trigger in this patient with a so heavy autoimmune clinical history?
5. Response: Thanks for this suggestion. We have revised the discussion and the conclusion sections in order to clearly explain the relationship of SARS-CoV-2 and AHA.
6. Comments on the Quality of English Language
6.Response: The grammatical errors in the English language have been revised and corrected.
Thank you for your work. I am sure that your suggestions and corrections improve the quality of the manuscipt.
Kind regards,
Marton Nemeth
Line 267"Note that autoimmune conditions without over AHA can affect hemostasis, too" over should be replaced by overt?
Response: Thanks for this suggestion. The text has been corrected in the revised manuscript.
Reviewer 2 Report
I have the following suggestions:
1. The search query is stated as "“(covid-19 OR SARS-CoV-2 OR coronavirus) AND haemophilia”; since 'haemophilia' is often also spelt 'hemophilia' authors should confirm that use of the word 'hemophilia' does not capture more citations.
2. page 3, line 98: "activated partial thromboplastin time (APTT) was not measured." this is incredible; why? especially given PT & fibrinogen were.
3. authors could briefly mention in discussion the additional 'risk' of AHA from immunization against COVID19; perhaps refer to/cite PubMed ID: 36055265, as this gives a nice summary.
Minor:
1. page 2, lines 45/46: "coronavirus diseases 2019 (COVID19)" should probably read "coronavirus disease 2019 (COVID19)"
2. page 6, lines 193-202. This text is repetitive of prior text, and can probably be deleted
Could be improved by review from a native English writer.
Author Response
Dear Sir/Madam,
Thank you for your detailed and valuable review of our paper. I hope that our answers are clear and sufficient for you.
I have the following suggestions:
1. The search query is stated as "“(covid-19 OR SARS-CoV-2 OR coronavirus) AND haemophilia”; since 'haemophilia' is often also spelt 'hemophilia' authors should confirm that use of the word 'hemophilia' does not capture more citations.
1. Response: Thank you for your insightful comments and suggestion. We have incorporated the suggestion accordingly.
3. page 3, line 98: "activated partial thromboplastin time (APTT) was not measured." this is incredible; why? especially given PT & fibrinogen were.
3 Response: Thanks for pointing out this important issue. Unfortunately, the APTT was not part of the basic hemostasis panel either in the ED or as part of the pre-operative setting. But after this and similar cases, the APTT, TT, and fibrinogen have been added to the original hemostasis panel which consisted of PT (or INR) in addition to the complete blood count with differentials.
4. authors could briefly mention in discussion the additional 'risk' of AHA from immunization against COVID19; perhaps refer to/cite PubMed ID: 36055265, as this gives a nice summary.
4. Response: Thanks for this valuable suggestion. We have revised the section of discussion and added new references as well.
Minor:
1. page 2, lines 45/46: "coronavirus diseases 2019 (COVID19)" should probably read "coronavirus disease 2019 (COVID19)"
Response: Thanks for the suggestion. It has been corrected in the revised manuscript.
2. page 6, lines 193-202. This text is repetitive of prior text, and can probably be deleted.
Response: The duplicated text has been removed in the revised manuscript.
Comments on the Quality of English LanguageCould be improved by review from a native English writer.
Response: Thanks for this important comment. The grammatical errors in the English language have been revised and corrected. And, the paper will be reviewed by a native English-speaking writer.
Thank you for your comments and suggestions. I am sure that these corrections will improve the quality of the manuscript
Kind regards,
Marton Nemeth